# CL²GEC: A Multi-Discipline Benchmark for Continual Learning in Chinese Literature Grammatical Error Correction

## Abstract

The growing demand for automated writing assistance in diverse academic domains highlights the need for robust Chinese Grammatical Error Correction (CGEC) systems that can adapt across disciplines. However, existing CGEC research largely lacks dedicated benchmarks for multi-disciplinary academic writing, overlooking continual learning (CL) as a promising solution to handle domain-specific linguistic variation and prevent catastrophic forgetting. To fill this crucial gap, we introduce **CL²GEC**, the first Continual Learning benchmark for Chinese Literature Grammatical Error Correction, designed to evaluate adaptive CGEC across multiple academic fields. Our benchmark includes 10,000 human-annotated sentences spanning 10 disciplines, each exhibiting distinct linguistic styles and error patterns. CL²GEC focuses on evaluating grammatical error correction in a continual learning setting, simulating sequential exposure to diverse academic disciplines to reflect real-world editorial dynamics. We evaluate large language models under sequential tuning, parameter-efficient adaptation, and four representative CL algorithms, using both standard GEC metrics and continual learning metrics adapted to task-level variation. Experimental results reveal that regularization-based methods mitigate forgetting more effectively than replay-based or naive sequential approaches. Our benchmark provides a rigorous foundation for future research in adaptive grammatical error correction across diverse academic domains.

## 1 Introduction

Chinese Grammatical Error Correction (CGEC) has evolved rapidly alongside the surge of large language models (LLMs) (Ye et al., 2025b; Qingsong et al., 2025) and intelligent writing assistants Li et al. (2022); Qiu et al. (2025); Li et al. (2024b); Zhang et al. (2025). Most existing CGEC benchmarks, however, are (1) learner or general domain oriented Zhang et al. (2022); Ma et al. (2022), and (2) evaluated in a static setting Xu et al. (2022); Ye et al. (2023b; 2024). As a result, they offer limited insight into how CGEC models behave in high-stakes professional domains, especially in scientific manuscripts where style, terminology, and error distribution vary markedly across disciplines.

We argue that real-world scientific writing introduces an under-explored challenge for CGEC: *continual domain adaptation* Wu et al. (2024); Guan et al. (2025). In practice, CGEC systems is expected to ingest papers from, e.g., Physics this month and Humanities next month, continually refining its internal knowledge without access to all past data. The threat of catastrophic forgetting Li et al. (2024a), widely studied in vision Shmelkov et al. (2017) and NLP tasks Shao & Feng (2022), has received almost no attention in CGEC, leaving an open question: *Can modern LLMs retain grammatical knowledge while sequentially adapting to new scientific disciplines?*

Addressing this question is crucial for three reasons. First, academia encompasses hundreds of sub-fields whose linguistic conventions differ in syntax and terminology, significantly challenging current LLMs. Second, annotation budgets are usually fragmented by discipline, making one-shot and full-data retraining impractical. Third, reliable cross-domain grammars underpin downstream tasks such as automatic reviewing Pang et al. (2025), plagiarism detection Quidwai et al. (2023), and literature summarization Li et al. (2024d).

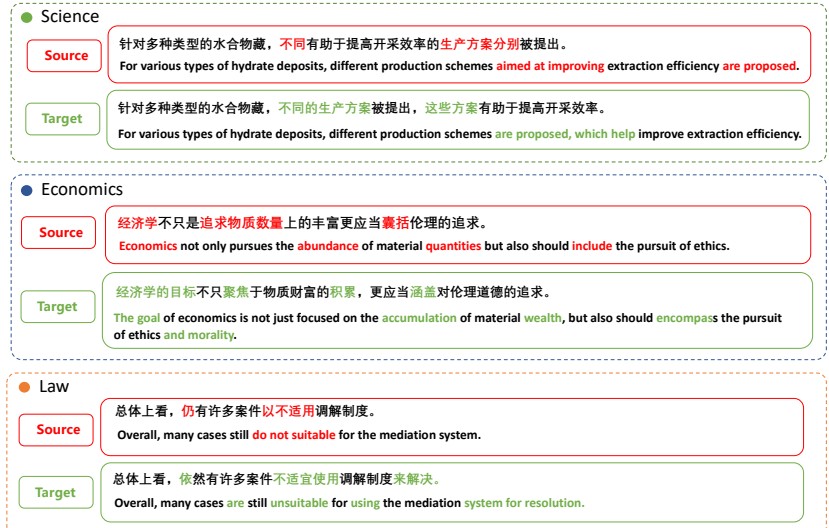

Figure 1: The correction examples in CL$^2$GEC.

Therefore, to systematically study CGEC under the context of continual learning (CL), we present **CL$^2$GEC**, the first Continual Learning benchmark for Chinese Literature Grammatical Error Correction. The CL$^2$GEC benchmark contains 10,000 sentences evenly sampled from 10 academic disciplines, each sentence paired with up to three independent human references. The corpus was curated from China National Knowledge Infrastructure (CNKI)[1], cleaned for copyright, and double-checked by professional editors to reflect authentic error patterns. We release both a canonical split (train/dev/test) and a sequence of 10 task partitions that simulate chronological arrival of disciplines, enabling controlled CL evaluation. CL$^2$GEC aims to set a new standard for evaluating and advancing lifelong grammatical error correction in the era of domain-diverse scientific communication.

Our proposed CL$^2$GEC allows researchers to probe a spectrum of model abilities, mainly including (1) in-domain grammatical accuracy, (2) cross-discipline transfer, and (3) resistance to catastrophic forgetting. Consequently, the benchmark fills a vital gap between generic CGEC datasets and real-world academic editing.

Empirically, we benchmark several representative continual learning strategies, including naive sequential fine-tuning, LoRA-based adaptation (Hu et al., 2021), and four CL algorithms (EWC (Kirkpatrick et al., 2017), GEM (Lopez-Paz & Ranzato, 2017), LwF (Li & Hoiem, 2016), OGD (Farajtabar et al., 2019)). Our comprehensive experiments reveal that while these methods significantly mitigate catastrophic forgetting compared to naive sequential approaches, the optimal strategies vary. We observe a nuanced impact of task ordering on knowledge retention and transfer, including unexpected interference between semantically related disciplines and a trade-off between precision and recall depending on the task sequence.

**We highlight our main contributions as follows:**

- We introduce CL$^2$GEC, the first large-scale and multi-discipline benchmark tailored to Chinese literature grammatical error correction in the context of continual learning.

- We devise task-specific CL metrics (Average Performance, Backward Transfer) adapted to CGEC, and provide a reproducible evaluation suite.

- We conduct extensive LLM experiments, revealing critical limitations of existing CL methods and establishing solid baselines for future work.

---

[1] https://www.cnki.net/

## 2 RELATED WORK

### 2.1 CHINESE GRAMMATICAL ERROR CORRECTION

Chinese grammatical error correction (CGEC) has developed from early sequence-to-sequence (Seq2Seq) models (Ye et al., 2023a; 2025a; Li et al., 2025), which model correction as a generation task. These approaches benefited from pretraining and syntactic priors but were mainly applied to general learner corpora.

With the advent of large language models (LLMs) (Kuang et al., 2025; Li et al., 2024c), recent work has explored their capabilities for CGEC (Wang et al., 2024; Xiao et al., 2024). Closed- and open-source LLMs have been evaluated through in-context learning and instruction tuning, showing improved fluency and generalization across error types. ScholarGEC (Kong & Wang, 2025) further investigates controllability in academic writing by combining error detection and correction within a multi-task framework.

While prior work focuses on enhancing model performance under static settings, relatively little attention has been paid to domain transfer or continual adaptation. Our CL$^2$GEC benchmark addresses this gap by providing a multi-discipline scientific dataset and evaluating models under domain-incremental settings, enabling more systematic assessment of generalization in CGEC.

### 2.2 CONTINUAL LEARNING FOR NLP

Continual learning (CL) aims to enable models to learn from sequentially arriving tasks while mitigating catastrophic forgetting (De Lange et al., 2021; Xing et al., 2024). Existing approaches are typically grouped into three categories: regularization-based, replay-based, and architectural-based.

**Regularization-based methods** introduce constraints on parameter updates to preserve knowledge of prior tasks. For example, Orthogonal Gradient Descent (OGD) (Farajtabar et al., 2019) enforces gradient orthogonality, Elastic Weight Consolidation (EWC) (Kirkpatrick et al., 2017) penalizes changes to parameters deemed important, and Gradient Episodic Memory (GEM) (Lopez-Paz & Ranzato, 2017) maintains an episodic buffer to avoid loss increases on earlier tasks. Learning without Forgetting (LwF) (Li & Hoiem, 2016) mitigates forgetting through distillation from previous model snapshots.

**Replay-based methods** explicitly revisit past knowledge by reusing stored samples or their approximations. The simplest variant is Experience Replay, which replays examples from a memory buffer, while more advanced approaches use generative replay or replay-based distillation such as SEEKR (He et al., 2024).

**Architectural-based methods** adapt the model structure to integrate new information, either by allocating task-specific components or expanding capacity. A representative example is Progressive Prompts (Razdaibiedina et al., 2023), which incrementally composes task-specific prompts into a composite representation.

While these approaches have been widely studied in NLP, grammatical error correction (GEC) presents unique challenges: it requires not only accuracy but also fluency and domain sensitivity. To our knowledge, continual learning has not been systematically explored for Chinese GEC (CGEC). Our work addresses this gap by introducing the first continual CGEC benchmark, enabling controlled evaluation across diverse academic disciplines.

## 3 CL$^2$GEC BENCHMARK

### 3.1 PROBLEM DEFINITION

**Grammatical Error Correction (GEC)** aims to transform an ungrammatical sentence $X = \{x_1, x_2, \ldots, x_T\}$ into its grammatically correct counterpart $Y = \{y_1, y_2, \ldots, y_{T'}\}$ while preserving the original semantics. Typically formulated as a sequence-to-sequence task, GEC models are

trained to minimize the negative log-likelihood of the corrected output:

$$\mathcal{L}_{\text{GEC}} = -\sum_{t=1}^{T'} \log P(y_t \mid Y_{<t}, X). \tag{1}$$

**Continual Learning (CL)** addresses the challenge of learning from a stream of tasks $\{\mathcal{D}_1, \ldots, \mathcal{D}_N\}$ without access to previous task data. In the supervised setting, each task $\mathcal{D}_t = \{(x_i^t, y_i^t)\}_{i=1}^{n_t}$ is presented sequentially, and the model is trained to accumulate knowledge over time while avoiding catastrophic forgetting. Let $f_\Theta$ be a model with parameters $\Theta$. The goal of CL is to optimize performance across all tasks:

$$\max_{\Theta} \sum_{t=1}^{N} \sum_{(x,y)\in\mathcal{D}_t} \log P_\Theta(y \mid x). \tag{2}$$

Evaluation in CL involves metrics such as *Average Performance* and *Backward Transfer*, which measure the model's ability to retain and transfer knowledge across tasks.

**The CL$^2$GEC Task**   We define CL$^2$GEC as a domain-specific Grammatical Error Correction (GEC) benchmark formulated under the continual learning (CL) paradigm. The dataset is composed of grammatically erroneous academic sentences collected from 10 distinct disciplines (e.g., law, medicine, philosophy), with each domain corresponding to a sequential task:

$$\mathcal{D}_t = \{(X_i^t, Y_i^t)\}_{i=1}^{n_t}. \tag{3}$$

Each pair consists of an erroneous sentence $X_i^t$ and its corrected version $Y_i^t$. Unlike typical GEC tasks trained on all data simultaneously, CL$^2$GEC is formulated as a continual learning benchmark where the model learns sequentially from each domain and only replays a small subset of previous data. This setup simulates real-world constraints such as limited storage and privacy, requiring the model to maintain performance across diverse domains without catastrophic forgetting.

## 3.2 BENCHMARK CONSTRUCTION

**Data Collection.**   We crawl full-text PDFs from the China National Knowledge Infrastructure (CNKI)[2], the largest Chinese academic repository. To capture broad domain diversity, we target 10 first-level disciplines: *Law, Management, Education, Economics, Science, History, Agriculture, Literature, Art*, and *Philosophy*. The dataset is designed to be highly diverse and multi-disciplinary, ensuring comprehensive coverage of academic writing. These 10 first-level disciplines are further subdivided into a total of 100 second-level disciplines, providing a granular representation of academic writing. For instance, the Agriculture discipline includes sub-disciplines such as Agricultural Resources and Environment (164 instances). The detailed breakdown of all disciplines and their instance counts can be found in the Table 1. For each discipline, we randomly sample 1,000 sentences, yielding a balanced corpus of 10,000 instances. This one-to-one ratio eliminates domain-size bias and guarantees that subsequent continual-learning curricula are not dominated by any single field. To ensure a standardized evaluation, we provide a canonical split for each discipline: 800 training examples, 100 development examples, and 100 test examples.

Table 1: Error Type Statistics by First-Level Discipline.

| Discipline | Total Errors | Word Omission | Word Misuse | Redundancy | Punctuation Errors | Sentence Blend | Ambiguity/Logic | Others |
|---|---|---|---|---|---|---|---|---|
| Education | 3,950 | 836 (21.2%) | 1,159 (29.3%) | 541 (13.7%) | 468 (11.8%) | 213 (5.4%) | 43 (1.1%) | 690 (17.5%) |
| Management | 3,215 | 845 (26.3%) | 1,232 (38.3%) | 457 (14.2%) | 379 (11.8%) | 172 (5.4%) | 48 (1.5%) | 82 (2.5%) |
| History | 2,890 | 688 (23.8%) | 911 (31.5%) | 454 (15.7%) | 352 (12.2%) | 177 (6.1%) | 36 (1.2%) | 272 (9.4%) |
| Law | 3,420 | 645 (18.9%) | 1,002 (29.3%) | 452 (13.2%) | 496 (14.5%) | 173 (5.1%) | 68 (2.0%) | 584 (17.1%) |
| Science | 4,105 | 930 (22.7%) | 1,288 (31.4%) | 579 (14.1%) | 442 (10.8%) | 218 (5.3%) | 59 (1.4%) | 589 (14.3%) |
| Philosophy | 3,215 | 624 (19.4%) | 1,171 (36.4%) | 503 (15.6%) | 414 (12.9%) | 208 (6.5%) | 56 (1.7%) | 239 (7.5%) |
| Economics | 3,450 | 751 (21.8%) | 1,127 (32.7%) | 517 (15.0%) | 431 (12.5%) | 201 (5.8%) | 61 (1.8%) | 362 (10.5%) |
| Agriculture | 2,980 | 773 (25.9%) | 953 (32.0%) | 412 (13.8%) | 386 (12.9%) | 161 (5.4%) | 44 (1.5%) | 251 (8.4%) |
| Literature | 2,715 | 595 (21.9%) | 1,031 (38.0%) | 401 (14.8%) | 367 (13.5%) | 151 (5.6%) | 42 (1.6%) | 128 (4.7%) |
| Art | 2,145 | 503 (23.5%) | 935 (43.6%) | 436 (20.3%) | 289 (13.5%) | 170 (7.9%) | 19 (0.9%) | 52 (2.4%) |

---

[2]https://www.cnki.net/

**Data Cleaning.** Because CNKI only provides PDF files, a dedicated preprocessing pipeline is required. The overall cleaning procedure is executed by a trained annotation team and guarantees that every remaining sentence is grammatically self-contained and suitable for correction.

1. **PDF → JSON conversion**. We convert each PDF into a structured JSON file that preserves sentence boundaries, section tags, and positional metadata. This machine-readable format facilitates downstream filtering and reproducibility.

2. **Section filtering**. Only the *abstract* and *main body* are retained. Other sections like references, acknowledgements are discarded. These sections contain the bulk of scientific exposition and therefore the majority of grammar-related errors relevant to writing assistance.

3. **Sentence segmentation**. The retained text is split into sentence-level units using *LTP* Che et al. (2021), enabling sentence-level GEC evaluation.

4. **Noise removal**. Inline citations, sub-titles, mathematical equations, tables, figures, and their captions are stripped. Eliminating non-linguistic tokens avoids misleading the error-detection models and prevents annotators from spending time on irrelevant content.

5. **Anonymisation**. All personal, institutional, and document identifiers are redacted to comply with privacy regulations and facilitate open release.

**Data Annotation.** Given the low error density in scholarly writing, fully manual annotation would be prohibitively expensive. We therefore adopt a human-in-the-loop strategy that combines automatic grammatical error detection, LLM pre-correction, human annotation, and expert validation.

1. **Automatic grammatical error detection**. 6 well-trained CGEC models (including GEC-ToR Omelianchuk et al. (2020) and fine-tuned Chinese-BART Shao et al. (2021)) are first applied to the cleaned corpus. Only sentences flagged *consistently* by *all* detectors are kept. This consensus voting filters out roughly 95% of grammatically correct lines, concentrating annotation effort on the 5% most error-prone candidates and dramatically cutting both LLM invocation and human labor.

2. **LLM pre-correction**. The shortlisted sentences are passed to GPT-4o Hurst et al. (2024), which produces a candidate correction for each error span. These machine suggestions serve as weak references, giving annotators a standardising correction style across workers.

3. **Human annotation**. We recruit senior undergraduates or graduates with majors that *match* the corresponding discipline, ensuring domain awareness. After extensive training on pilot samples, annotators correct each sentence while consulting the detector output and GPT-4o suggestions. Every instance is independently revised by at least two annotators, which both improves recall and exposes stylistic alternatives.

4. **Expert validation**. Domain experts (including the paper authors) perform 100% manual review of the double-annotated data. They refine erroneous edits, reconcile conflicts, and may add supplementary references when multiple acceptable rewrites exist. The outcome is a high-precision, multi-reference gold annotation set.

This multi-stage pipeline maximises annotation quality *and* cost-effectiveness: automatic error detection minimises wasted effort, LLM pre-correction accelerates human editing, dual annotation guarantees inter-annotator agreement, and expert review delivers publication-grade reliability.

### 3.3 EVALUATION PROCEDURE AND METRICS

We evaluate model performance in a continual learning setting using both standard GEC metrics and continual learning metrics.

### 3.3.1 EVALUATION PROTOCOL

Let $\{T_1, \ldots, T_N\}$ denote the sequence of $N = 10$ tasks, corresponding to our ten academic disciplines. In this continual learning setup, models are trained sequentially on discipline-specific training sets. To assess robustness to domain shift, we consider two curricula: (i) a semantically ordered sequence and (ii) a randomly shuffled sequence.

After learning each task $T_i$, the model is evaluated on all tasks from $T_1$ up to $T_i$. We record the following scores: (1) $Q_{i,i}$, the performance on the current task $T_i$ immediately after training; (2) $Q_{i,j}$ ($j < i$), the performance on a past task $T_j$ after training on $T_i$; and (3) $Q_{N,j}$, the final performance on each task $T_j$ after completing the entire sequence of $N$ tasks.

### 3.3.2 STANDARD GEC METRICS

We evaluate grammatical error correction (GEC) performance using the ChERRANT scorer (Zhang et al., 2022), which extends ERRANT to Chinese by performing character-level alignment and edit classification. For each evaluation result $Q_{i,j}$, ChERRANT computes Precision (P), Recall (R), and $F_{0.5}$ by classifying edits as true positives, false positives, or false negatives.

To summarize performance across tasks, we compute averages independently for each metric $M \in \{P, R, F_{0.5}\}$:

$$\overline{Q}^{(M)} = \frac{1}{N} \sum_{j=1}^{N} Q_{N,j}^{(M)}. \tag{4}$$

Here $Q_{N,j}^{(M)}$ denotes the final score on task $T_j$ under metric $M$. Importantly, $F_{0.5}$ is averaged directly across tasks rather than recomputed from the averaged P and R.

### 3.3.3 CONTINUAL LEARNING METRICS

To capture retention and overall competence under sequential training, we adopt two standard continual learning metrics, computed separately for each $M \in \{P, R, F_{0.5}\}$.

**Backward Transfer (BWT).** BWT measures the average change in performance on past tasks from immediately after training to the end of the sequence. Negative values indicate forgetting, while non-negative values indicate retention or positive transfer:

$$\text{BWT}^{(M)} = \frac{1}{N-1} \sum_{j=1}^{N-1} \left( Q_{N,j}^{(M)} - Q_{j,j}^{(M)} \right). \tag{5}$$

**Average Task Performance (AvgPerf).** AvgPerf measures the model's overall GEC ability across all tasks after completing the full training sequence:

$$\text{AvgPerf}^{(M)} = \frac{1}{N} \sum_{j=1}^{N} Q_{N,j}^{(M)}. \tag{6}$$

## 4 EXPERIMENTS

### 4.1 EXPERIMENTAL SETTINGS

**Backbones.** All experiments use Qwen2.5-7B-Instruct (Qwen et al., 2025) and Llama3-8B-Instruct (Llama Team, 2024) as backbone models, chosen for their strong Chinese-language performance and robust instruction-following capabilities.

**Task Sequences and Evaluation.** To comprehensively assess the impact of task order on continual learning performance, we conducted experiments using two distinct training sequences:

1. **Randomized Order**: The 10 academic disciplines are presented in a randomly shuffled order. To ensure robustness and account for variance, this process is repeated across 5 different random permutations. We report the average results across these runs.

2. **Semantically Similar Order**: The tasks are arranged based on their semantic similarity. This sequence simulates a smoother domain transition and is used to investigate the effect of gradual domain shift on catastrophic forgetting.The definition and computation of similarity, as well as the resulting task order, are detailed in the Appendix.

Table 2: Main Results of CL Strategies on $CL^2GEC$, Random (Rnd) vs. Semantic (Sem) Order.

| Model | Strategy | GEC (Rnd) | | | GEC (Sem) | | | AvgPerf (Rnd) | | | AvgPerf (Sem) | | | BWT (Rnd) | | | BWT (Sem) | | |
|---|---|---|---|---|---|---|---|---|---|---|---|---|---|---|---|---|---|---|---|
| | | P | R | $F_{0.5}$ | P | R | $F_{0.5}$ | P | R | $F_{0.5}$ | P | R | $F_{0.5}$ | P | R | $F_{0.5}$ | P | R | $F_{0.5}$ |
| Qwen2.5 7B-Instruct | SeqFT | 59.25 | 10.71 | 29.91 | 52.10 | 12.61 | 31.08 | 50.92 | 11.97 | 29.57 | 46.70 | 14.84 | 31.18 | **8.13** | -1.20 | -0.40 | 0.95 | -0.13 | 0.63 |
| | LoRA | 65.42 | 13.00 | 35.54 | 64.18 | 13.03 | 34.83 | 62.80 | 11.33 | 32.02 | 61.70 | 12.92 | 33.94 | 4.54 | 1.61 | 4.01 | 1.77 | **1.55** | **3.00** |
| | Replay | 58.78 | 11.49 | 31.13 | 47.04 | 11.22 | 26.90 | 50.85 | 12.58 | 29.93 | 48.00 | 13.89 | 30.89 | 5.75 | -1.73 | -0.65 | -4.31 | -1.64 | -4.17 |
| | EWC | 67.34 | 13.00 | 35.52 | 64.26 | 12.97 | 34.76 | **64.88** | 11.40 | 32.11 | 61.70 | 12.93 | 33.94 | 4.44 | 1.60 | 4.00 | 1.94 | 1.40 | 2.77 |
| | GEM | 67.33 | 13.10 | **35.65** | **64.34** | 13.00 | 34.82 | 64.86 | 11.42 | 32.17 | 61.77 | 12.93 | 33.96 | 4.41 | **1.70** | **4.11** | **1.99** | 1.45 | 2.84 |
| | LwF | 62.86 | **13.22** | 34.89 | 63.73 | 12.98 | 34.75 | 58.60 | **12.84** | **33.36** | 61.36 | 13.84 | 35.24 | 3.10 | 0.01 | 0.69 | 0.95 | 0.50 | 0.96 |
| | OGD | **67.78** | 12.30 | 34.44 | 62.77 | **14.43** | **36.53** | 63.15 | 13.38 | 34.97 | **62.07** | **15.18** | **37.08** | 4.85 | -1.32 | -0.81 | -1.71 | 0.78 | 0.61 |
| LLaMA3 8B-Instruct | SeqFT | 45.40 | 9.10 | 24.14 | 45.60 | 9.98 | 26.01 | 35.15 | 10.73 | 22.52 | 36.56 | 11.03 | 24.00 | 6.11 | -2.02 | -1.32 | 4.03 | -0.24 | 0.78 |
| | LoRA | 64.21 | 11.11 | 31.62 | 56.80 | 12.98 | 33.03 | 58.93 | 11.36 | 30.98 | 56.32 | 12.85 | 32.32 | 7.51 | -0.22 | 0.64 | -1.67 | 1.97 | 2.62 |
| | Replay | 37.31 | 10.07 | 23.70 | 34.32 | 9.56 | 22.17 | 30.00 | 10.39 | 20.79 | 27.90 | 10.77 | 20.46 | 7.41 | -0.60 | **2.50** | **7.73** | -0.52 | 2.64 |
| | EWC | 63.82 | 11.00 | 31.34 | 57.06 | 13.11 | 33.29 | 58.80 | 11.33 | 30.91 | 56.35 | **12.91** | **32.38** | 7.63 | -0.22 | 0.67 | -2.34 | 1.96 | 2.55 |
| | GEM | **64.46** | 11.12 | 31.71 | 57.94 | 13.16 | 33.53 | **58.97** | 11.35 | 30.99 | 56.28 | 12.82 | 32.26 | **8.13** | -0.12 | 0.96 | -0.67 | 2.19 | 3.16 |
| | LwF | 59.75 | 11.18 | 30.98 | **59.79** | 10.18 | 29.80 | 57.24 | 11.60 | 30.86 | **58.99** | 11.56 | 31.29 | 3.04 | -0.43 | 0.06 | 0.09 | 0.87 | 1.53 |
| | OGD | 60.37 | **12.89** | **33.54** | 57.99 | **13.42** | **33.89** | 57.12 | 12.92 | **32.74** | 56.56 | 12.74 | 32.37 | 3.15 | **-0.05** | 0.49 | -1.07 | **2.30** | **2.92** |

Models are sequentially adapted to the 10 disciplines according to these task sequences. For each run, evaluation is conducted after training on each task, following the procedure outlined in the evaluation section. The final results for the randomized order are averaged over the 5 permutations. Evaluation is performed using official CGEC scoring scripts.

## 4.2 CONTINUAL LEARNING METHODS

We investigate the performance of various continual learning methods applied to the domain-specific GEC task. Given that our task requires adapting a large language model to a series of distinct yet related domains, we focus on a strategy combining Parameter-Efficient Tuning (PET) with established continual learning algorithms. To this end, we benchmark the following four categories of adaptation strategies:

- **Sequential Finetuning (SeqFT)**: A naive baseline where the model is trained on each task in sequence without any specific mechanism to retain prior knowledge. This approach provides a lower bound for performance and highlights the problem of catastrophic forgetting.

- **Parameter-Efficient Tuning (LoRA)**: We apply Low-Rank Adaptation with rank 8. This serves as a lightweight adaptation approach.

- **Replay-based Methods**: To mitigate forgetting, we implement experience replay by retaining 2%, 5%, or 10% of training data from previous tasks.

- **Continual Learning Algorithms**: For our single-domain GEC task, we combine Parameter-Efficient Tuning (LoRA) with a set of representative continual learning algorithms to achieve superior results. We evaluate four such approaches: **EWC** (Elastic Weight Consolidation) (Kirkpatrick et al., 2017), which regularizes important parameters; **LwF** (Learning without Forgetting) (Li & Hoiem, 2016), which uses knowledge distillation; **GEM** (Gradient Episodic Memory) (Lopez-Paz & Ranzato, 2017), which constrains gradient updates; **OGD** (Orthogonal Gradient Descent) (Farajtabar et al., 2019), which minimizes task interference through orthogonal updates.

## 5 ANALYSIS

Our analysis of experiments at the $CL^2GEC$ benchmark reveals several critical insights into the efficacy of continual learning (CL) strategies for large language models that perform Grammatical Error Correction (GEC). We discuss the three dimensions (GEC, AvgPerf, and BWT) and the impact of the two task orders. The results highlight not only the performance of different methods but also the crucial impact of model choice and task ordering.

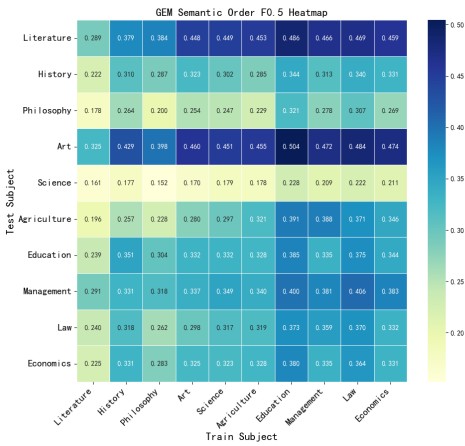

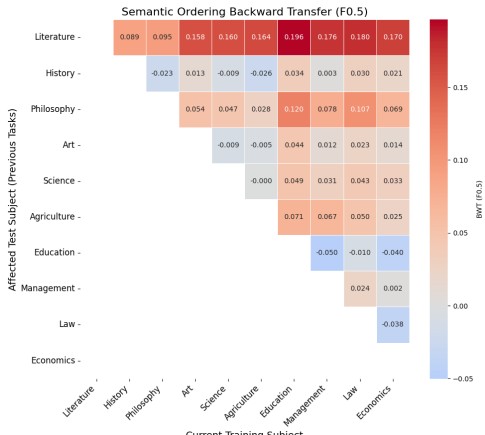

Figure 2: GEM Semantic Order F0.5 Heatmap.

Figure 3: Backward Transfer (F0.5) with Semantic Ordering.

## 5.1 OVERALL PERFORMANCE

Across both backbones and task orders, continual learning (CL) strategies clearly outperform sequential fine-tuning and generally surpass LoRA, while Replay remains unreliable. This highlights the particular sensitivity of grammatical error correction (GEC) to catastrophic forgetting and the necessity of mechanisms that explicitly preserve previously acquired knowledge. Among the backbones, Qwen2.5-7B-Instruct consistently achieves higher scores than LLaMA3-8B-Instruct, suggesting that multilingual pretraining provides stronger inductive biases for this task; nevertheless, the relative ordering of methods remains stable across models, underscoring the robustness of the observed trends. LoRA serves as a strong baseline by constraining updates to a low-rank subspace, which alleviates the most severe forgetting, but its stability decreases under semantically ordered tasks, showing that parameter-efficient adaptation alone cannot fully prevent drift. CL methods systematically close this gap by directly addressing task interference, yielding more consistent performance across domains and curricula.

## 5.2 CONTINUAL LEARNING STRATEGIES

While CL methods uniformly outperform the baselines, their strengths diverge systematically according to algorithmic design.

**Projection-based methods (OGD) emphasize forward plasticity.** By enforcing gradient orthogonality, OGD effectively incorporates new error patterns from diverse domains and achieves the strongest overall performance. This comes at the expense of weaker backward transfer, reflecting its bias toward adaptability rather than long-term retention.

**Constraint-based methods (GEM, EWC) prioritize stability.** GEM attains the strongest backward transfer, consistent with its objective of constraining updates to protect earlier tasks—a natural fit for GEC, where many grammatical structures recur across domains. EWC provides a more balanced trade-off, maintaining competitive final performance while offering robust retention.

**Distillation-based methods (LwF) provide moderate but consistent gains.** By distilling predictions from earlier tasks, LwF reduces forgetting without requiring memory storage. It consistently improves over LoRA, though generally trails OGD and GEM in overall performance.

**Replay is less effective in this setting.** A small memory buffer fails to capture the structural diversity of GEC errors, leading to unstable performance across task orders. Constraint- or projection-based methods appear more effective than raw memory in this benchmark.

## 5.3 Impact of Task Order

Task ordering exerts a nuanced yet systematic influence on continual learning outcomes. For average performance, semantic curricula generally improve recall and $F_{0.5}$ across most strategies, but often lead to a decline in precision. This pattern is also observable in GEM, EWC, and Replay, suggesting that exposure to related tasks encourages the model to generalize more broadly, increasing coverage (recall) at the expense of specificity (precision). This trade-off may reflect greater ambiguity in correction boundaries across similar domains. By contrast, random orderings present a more diverse distribution of topics, which may sharpen task boundaries, yielding higher precision but weaker cross-task generalization.

The effect on backward transfer (BWT) is more model-dependent. For Qwen2.5-7B-Instruct, semantic ordering often reduces BWT, indicating that consecutive exposure to similar domains may amplify interference and limit the reuse of prior knowledge. In contrast, LLaMA3-8B-Instruct frequently shows improved BWT under semantic order—for instance with GEM and LoRA—suggesting that it benefits from redundancy across related tasks, possibly consolidating representations that were less well established during pretraining. These results highlight that task-order effects are not universal but are mediated by both the model's inductive biases and the mechanisms of the CL strategy.

## 5.4 Case Study: GEM under Different Task Orders

To better understand the dynamics of continual learning, we conduct a detailed case study of the GEM strategy using $F_{0.5}$ and BWT heatmaps (Figures 2 and 4). These visualizations provide a fine-grained view of how knowledge is preserved or overwritten as training progresses across tasks. For clarity, we present only the semantic-order results in the main text; the corresponding random-order heatmaps are included in the appendix.

**$F_{0.5}$ performance stability.** The heatmap in Figure 2 tracks $F_{0.5}$ scores for each test subject (y-axis) after successive training tasks (x-axis, ordered from left to right). GEM maintains strong and relatively stable performance across the sequence, confirming its effectiveness in mitigating catastrophic forgetting. Compared to random ordering (Appendix, Figure **??**), semantic ordering yields smoother performance transitions, indicating that related tasks reinforce one another and support more predictable accumulation of knowledge.

**Backward transfer dynamics.** Figure 4 shows the BWT matrix, where the x-axis denotes the current training task and the y-axis denotes the previously learned test task. Semantic ordering promotes strong positive transfer among related domains—for example, `Literature` consistently benefits from subsequent training on other humanities disciplines, with BWT values reaching as high as 0.196. This illustrates how semantically structured curricula can leverage domain synergies to strengthen prior knowledge. At the same time, semantic ordering exhibits localized vulnerability: tasks less aligned with the curriculum suffer persistent negative BWT, indicating systematic forgetting. In contrast, random ordering (Appendix, Figure 5) yields more scattered and less severe negative BWT values, reducing the likelihood of any single task being consistently overwritten, though at the cost of weaker and less predictable positive transfer.

## 6 Conclusion

We introduced **$CL^2GEC$**, the first continual learning benchmark for Chinese grammatical error correction (CGEC) in academic writing. $CL^2GEC$ simulates domain-incremental learning through a 10-discipline corpus of 10,000 human-annotated sentences, enabling sequential training and fine-grained evaluation of forgetting, adaptation, and transfer. We defined tailored evaluation protocols and benchmarked strong baselines using parameter-efficient tuning and four representative CL algorithms. Results show that regularization- and projection-based methods outperform sequential fine-tuning and replay, though performance varies with task order and model backbone. We hope $CL^2GEC$ provides a foundation for future work on adaptive GEC systems and inspires the development of lifelong writing assistants capable of generalizing across academic domains.

# 7 LIMITATIONS

**Reliance on a Specific Type of Continual Learning.** This work primarily evaluates regularization-based continual learning methods like OGD and GEM. While these methods show strong performance, other approaches such as memory-augmented or meta-learning-based methods might offer complementary benefits. Future research could explore these alternatives for a more comprehensive understanding of continual learning in CGEC.

**Generalization to Other Languages.** Our experiments focus on Chinese GEC, and the methods proposed may not directly generalize to other languages with different grammatical structures. Further research is needed to assess the applicability of $CL^2GEC$ to languages other than Chinese.

**Evaluation on Limited Model Architectures.** We evaluate two large language models, Qwen2.5-7B-Instruct and LLaMA3-8B-Instruct. However, other model architectures, including smaller models or domain-specific models, may yield different results. Expanding the evaluation to include a wider range of architectures would provide a broader perspective on the effectiveness of continual learning methods.

# 8 ETHICS STATEMENT

In this paper, we introduce the $CL^2GEC$ benchmark, which is constructed from a custom-curated dataset. We have carefully detailed the collection, preprocessing, and annotation processes to ensure that no unethical behavior or infringement occurred during the dataset construction. To comply with ethical standards, we focus on data anonymization, desensitization, and the removal of any potentially harmful or biased content. The texts used in our dataset are sourced from publicly available academic materials, ensuring that the research tasks and directions proposed do not cause harm to society.

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

# A   APPENDIX

## A.1   DATA BREAKDOWN AND SECONDARY DISCIPLINES

As described in the main text, the $CL^2GEC$ benchmark is composed of 10 primary disciplines, each containing a number of secondary disciplines. Since the total number of secondary disciplines is large, we focus on the most prominent ones for visualization. Figures 6 and 7 show the distribution of the top 10 most frequent secondary disciplines within each primary discipline. To visualize this distribution, a stacked bar chart is provided, illustrating the count and proportion of these top 10 secondary disciplines within each primary discipline.

### A.1.1   RANDOMIZED ORDER

The 10 academic disciplines are presented in a randomly shuffled order. To ensure robustness and account for variance, this process is repeated across 5 different random permutations. We report the average results across these runs.

### A.1.2   SEMANTICALLY SIMILAR ORDER

The tasks are arranged based on the semantic similarity between the disciplines. The resulting order is designed to reflect a smoother transition between related domains, thus investigating how gradual shifts between similar tasks affect the model's ability to retain knowledge. The semantic similarity between disciplines is computed using sentence embeddings generated by the `SentenceTransformer` model. The average similarity matrix between the disciplines is presented in Table 4.

**Semantic Similarity Computation**   We compute the semantic similarity between disciplines by first encoding each discipline's sentences into dense vector representations (embeddings) using the `SentenceTransformer` model and then averaged to produce a single vector for each discipline. We compute the cosine similarity between all pairs of discipline embeddings to measure how similar they are to each other in a semantic space.

The formula for cosine similarity between two vectors $\mathbf{v_1}$ and $\mathbf{v_2}$ is:

$$\text{Cosine Similarity} = \frac{\mathbf{v_1} \cdot \mathbf{v_2}}{\|\mathbf{v_1}\|\|\mathbf{v_2}\|}$$

This cosine similarity value ranges from -1 (completely dissimilar) to 1 (identical). The results of this computation are presented in Table 4.

**Conclusion of Task Ordering**   Based on the semantic similarity matrix, we categorize the tasks into three groups for training order:

- Literature, History, Philosophy, and Art Group
- Education, Management, Law, and Economics Group
- Science, Agriculture Group

## A.2   REPLAY STRATEGY

We further analyze the replay strategy by varying the buffer size (2%, 5%, 10%) on both Qwen2.5-7B-Instruct and LLaMA3-8B-Instruct (Table 3). The results reveal that replay does not scale monotonically with buffer size; instead, its effectiveness peaks at small to medium buffers and is strongly model-dependent.

Qwen2.5-7B-Instruct achieves its best performance at a 5% buffer under random order, but both GEC and AvgPerf decline when the buffer is enlarged to 10%. Moreover, backward transfer (BWT) remains consistently negative, with particularly sharp degradation under semantic order, suggesting that replay alone is insufficient to mitigate forgetting for Qwen2.5-7B-Instruct and that larger buffers may even introduce redundancy or noise. By contrast, LLaMA3-8B-Instruct benefits more from

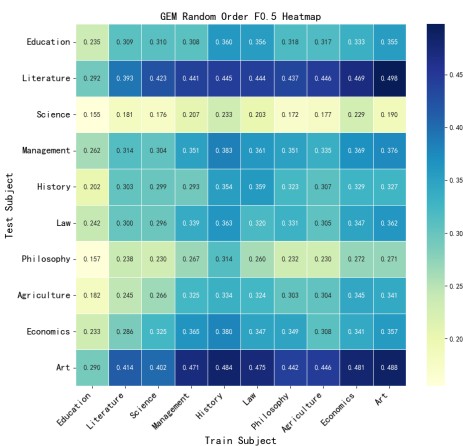
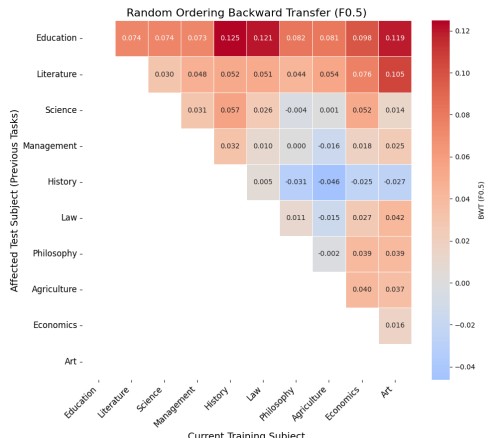

Figure 4: Backward Transfer (F0.5) with Random Ordering.

Figure 5: Backward Transfer (F0.5) with Random Ordering.

replay, with its strongest GEC and AvgPerf observed at 2% buffer, and BWT peaking around 5% before dropping again at 10%. This indicates that smaller buffers are more effective for consolidating past knowledge, while excessively large buffers blur task boundaries and reduce retention efficiency.

Table 3: Replay Results on $CL^2$GEC Benchmark.

| Model | Buffer | GEC (Rnd) | | | GEC (Sem) | | | AvgPerf (Rnd) | | | AvgPerf (Sem) | | | BWT (Rnd) | | | BWT (Sem) | | |
|---|---|---|---|---|---|---|---|---|---|---|---|---|---|---|---|---|---|---|---|
| | | P | R | $F_{0.5}$ | P | R | $F_{0.5}$ | P | R | $F_{0.5}$ | P | R | $F_{0.5}$ | P | R | $F_{0.5}$ | P | R | $F_{0.5}$ |
| **Qwen2.5 7B-Instruct** | 2% | 56.69 | 11.15 | 30.02 | **50.56** | 12.15 | **30.12** | 49.56 | 12.27 | 29.25 | 47.94 | **14.44** | **31.27** | 5.39 | -1.26 | **-0.61** | **-0.54** | -0.68 | -0.73 |
| | 5% | **58.78** | **11.49** | **31.13** | 47.04 | 11.22 | 26.90 | **50.85** | **12.58** | **29.93** | 48.00 | 13.89 | 30.89 | **5.75** | -1.73 | -0.65 | -4.31 | -1.64 | -4.17 |
| | 10% | 53.45 | 11.48 | 29.92 | 47.51 | **12.18** | 29.13 | 49.45 | 12.24 | 29.50 | **48.56** | 13.01 | 30.21 | 2.40 | **-1.13** | -0.64 | -2.97 | **-0.67** | **-0.59** |
| **LLaMA3 8B-Instruct** | 2% | **45.24** | 8.60 | **23.71** | **43.47** | 9.33 | **24.62** | **33.02** | 10.37 | **21.68** | **34.71** | **11.00** | **23.50** | **13.36** | -2.06 | 1.32 | **9.20** | -1.10 | 1.31 |
| | 5% | 37.31 | **10.07** | 23.70 | 34.32 | **9.56** | 22.17 | 30.00 | **10.39** | 20.79 | 27.90 | 10.77 | 20.46 | 7.41 | **-0.60** | **2.50** | 7.73 | **-0.52** | **2.64** |
| | 10% | 37.25 | 8.85 | 22.06 | 33.54 | 8.4 | 20.62 | 29.36 | 9.41 | 19.93 | 30.61 | 10.02 | 21.01 | 9.38 | -1.03 | 2.11 | 4.26 | -0.94 | 0.74 |

## A.3 CASE STUDY

We provide a case study of the $CL^2$GEC benchmark across 10 disciplines. The results are visualized in Figure 8 9 10 11 12, where each discipline is represented by a selection of annotated cases, showing how the benchmark performs across different academic domains.

| Discipline | Literature | History | Philosophy | Education | Law | Science | Agriculture | Economics | Management | Art |
|---|---|---|---|---|---|---|---|---|---|---|
| **Literature** | 1.0000 | 0.2115 | 0.2130 | 0.1506 | 0.1418 | 0.0493 | 0.0334 | 0.1185 | 0.1406 | 0.1923 |
| **History** | 0.2115 | 1.0000 | 0.2345 | 0.2252 | 0.2059 | 0.0872 | 0.0846 | 0.1733 | 0.2139 | 0.1769 |
| **Philosophy** | 0.2130 | 0.2345 | 1.0000 | 0.2024 | 0.2388 | 0.0923 | 0.0792 | 0.2173 | 0.1999 | 0.1553 |
| **Education** | 0.1506 | 0.2252 | 0.2024 | 1.0000 | 0.2324 | 0.1230 | 0.1340 | 0.1869 | 0.2988 | 0.1224 |
| **Law** | 0.1418 | 0.2059 | 0.2388 | 0.2324 | 1.0000 | 0.1036 | 0.1019 | 0.2111 | 0.2266 | 0.1223 |
| **Science** | 0.0493 | 0.0872 | 0.0923 | 0.1230 | 0.1036 | 1.0000 | 0.2022 | 0.1449 | 0.1523 | 0.0758 |
| **Agriculture** | 0.0334 | 0.0846 | 0.0792 | 0.1340 | 0.1019 | 0.2022 | 1.0000 | 0.1548 | 0.1508 | 0.0338 |
| **Economics** | 0.1185 | 0.1733 | 0.2173 | 0.1869 | 0.2111 | 0.1449 | 0.1548 | 1.0000 | 0.2093 | 0.0915 |
| **Management** | 0.1406 | 0.2139 | 0.1999 | 0.2988 | 0.2266 | 0.1523 | 0.1508 | 0.2093 | 1.0000 | 0.1286 |
| **Art** | 0.1923 | 0.1769 | 0.1553 | 0.1224 | 0.1223 | 0.0758 | 0.0338 | 0.0915 | 0.1286 | 1.0000 |

Table 4: Semantic Similarity Matrix between Disciplines

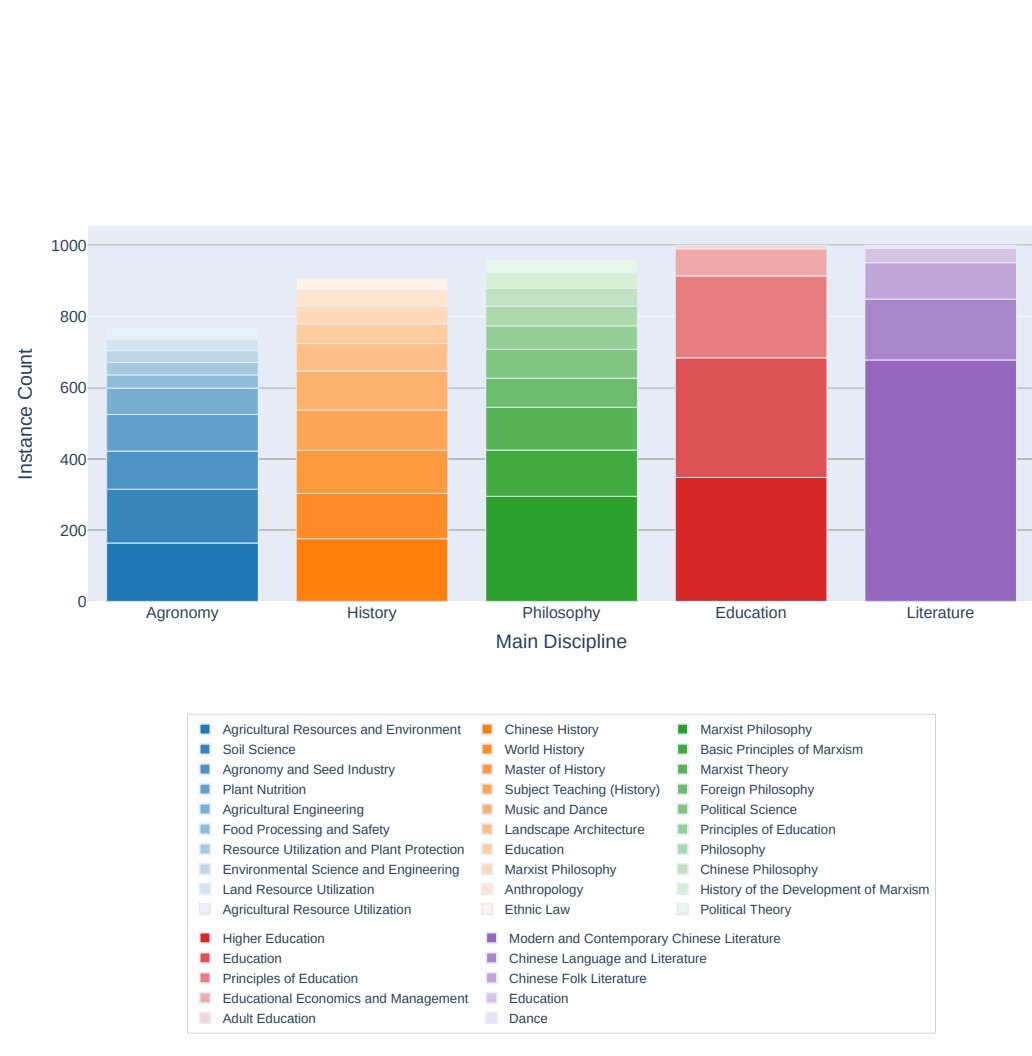

Figure 6: Discipline-wise Distribution of Top 10 Sub-disciplines (Part 1)

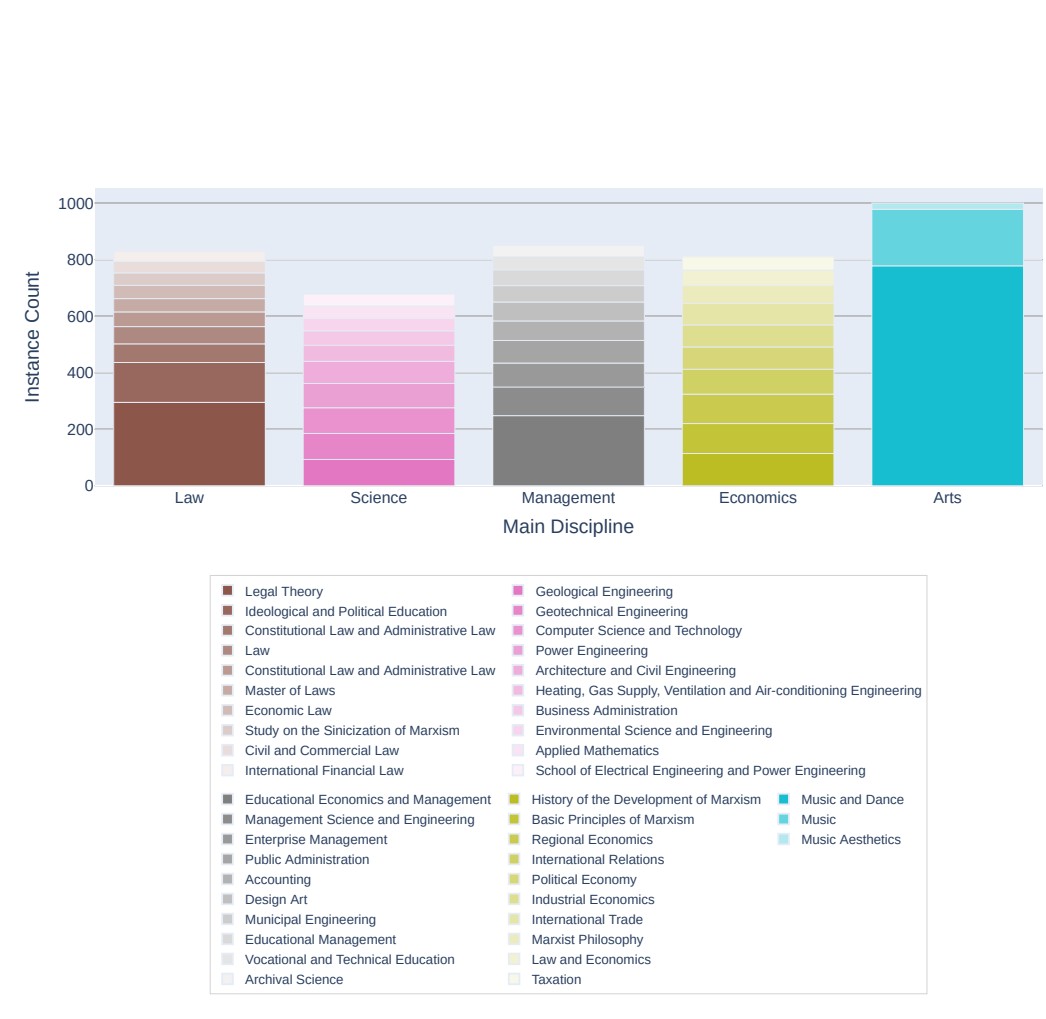

Figure 7: Discipline-wise Distribution of Top 10 Sub-disciplines (Part 2)

**1. 理学**

```
{
  "input": "当气体中所含溶质存有浓度差，这类浓度的不平衡态将会引发物质转移，研究者把这类现象称之为扩散。",

  "output": "当气体中所含溶质存有浓度差，这类浓度的不平衡态将会引发物质转移，研究者把这类现象称之为扩散。"
},
{
  "input": "近年来，BiOI 作为较为新鲜的光催化材料，因其独特的层状结构和新颖的光学性能而被广泛研究。",

  "output": "近年来，作为一种较为新鲜的光催化材料，BiOI 因其独特的层状结构以及新颖的光学性质受到了广泛的研究。"
}
```

**2. 法学**

```
{
  "input": "在研究方法上，实证研究方法的类型包括社会调查方法、历史研究方法、比较研究方法、逻辑分析方法、语义分析方法。",

  "output": "在研究方法上，实证研究方法包括社会调查方法、历史研究方法、比较研究方法、逻辑分析方法和语义分析方法。"
},
{
  "input": "社科法学除了需要在自身的方法论研究、研究内容上做出反思与改进外，社科法学要想在学术界作为一个"学派"走得更远，就必须在组织上形成自己的学术共同体。",

  "output": "社科法学除了需要在自身的方法论研究、内容上做出反思与改进外，社科法学要想在学术界作为一个"学派"走得更远，就必须在组织上形成自己的学术共同体。"
}
```

Figure 8: Examples of CL$^2$GEC.

**3. 管理学**

{

"input": "因此，为进一步提升学位论文的质量和水平，在学位论文选题中，既要考虑研究热点，也要考虑教育经济与管理学科范围内的本质问题、核心问题，而不能一味地去追求热点，只有这样研究成果才会被更多的人去关注，认可。",

"output": "所以，为了进一步提高学位论文的质量和水平，在选择学位论文题目时，既要关注研究热点，又要思考教育经济与管理学科范围内的本质问题、核心问题，不能盲目追求热点。这样，研究成果才会得到更多人的关注和认可。"

},

{

"input": "本文研究的意义在于探究式学习法这一较新的学习方式，可以使学生在探究和学习的过程中提高思考能力、学习能力、创新能力、协作能力，并对老师和学生之前的关系有着促进作用。",

"output": "本文研究的意义在于，探究式学习法作为一种较新的学习方式，能够让学生在探究和学习的过程中，提升思考能力、学习能力、创新能力和协作能力，并且对师生之间的关系起到促进作用。"

}

**4. 教育学**

{

"input": "高等教育学硕士研究生这样踏踏实实、勤勤恳恳的做学术研究、撰写论文，研究生三年实践扎扎实实学习积累，毕业前夕总能得到学术方面的"累累硕果"。 ",

"output": "高等教育学硕士研究生这样求真务实、兢兢业业地做学术研究、撰写论文，三年时间踏踏实实学习积累，毕业前夕就会得到学术方面的"累累硕果"。"

},

{

"input": "因为大学生是极具个性化的群体，因此大学生的发展表现出多样化，并极具个性差异，大学对于学生的管理要遵循学生的成长和发展规律。",

"output": "大学生是极具个性化的群体，表现出多样化并极具个性差异的特点，因此大学对于学生的管理要遵循学生的成长和发展规律。"

}

Figure 9: Examples of CL$^2$GEC.

**5. 经济学**

{

  "input": "总揽历史长河，经济增长理论的发展共经历了三个不同的阶段，分别是古典增长理论，新古典增长理论和新增长理论。",

  "output": "纵览历史长河，经济增长理论的发展共经历了三个不同的阶段，分别是古典增长理论、新古典增长理论和新增长理论。"

},

{

  "input": "增加的资本要雇佣更多的劳动，如果此时劳动者不能从人口更多的国家转移过来，这种资本增加的趋势就会大大的提高劳动的市场价格。",

  "output": "增加的资本要雇佣更多的劳动，如果此时劳动者不能从人口更多的国家转移过来，这种资本增加的趋势就会大大的提高劳动的市场价格。"

}

**6. 历史学**

{

  "input": "学界大体认为，二战前为传统的'汉学研究'，二战后则为现代的'中国学研究'，两者都致力于中国语言、历史和文化的研究。",

  "output": "学界大体认为，二战前为传统的"汉学研究"，二战后则为现代的"中国学研究"，两者都致力于中国语言、历史和文化的研究。"

},

{

  "input": "二是对历史研学旅行课程反思的探究。当学生历经了一次完整的研学课程后，对于历史研学课程有切身体会，是否学到了、学到了什么等可以作为探究的范例，可以在回顾知识的同时加深对知识的理解，从而在以后的课程学习中，达到"顿悟"的境界；三是对旁支学科的探究，达到跨学科学习。",

  "output": "二是对历史研学旅行课程反思的探究。当学生经历了一次完整的研学课程后，对于历史的研学课程有了切身体验。学生是否学到、学到了什么等可以作为探究的范例。可以在回顾知识的同时加深对知识的理解，从而在以后的课程学习中，达到"顿悟"的境界。三是对旁支学科的探究，达到跨学科学习。"

}

Figure 10: Examples of CL$^2$GEC.

### 7. 农学

```
{

  "input": "由于土壤养分数据获取成本较高，需要数据量大，更新困难，应进一步探讨
  实时的养分数据更新方法，降低数据更新周期，提高其实时性。",

  "output": "由于土壤养分数据获取成本较高、需要数据量大且更新困难，应进一步探
  讨实时的养分数据更新方法，以缩短数据更新周期，并提高其实时性。"

},
{

  "input": "在花生上以及小麦上的研究也表现出类似的结果，施用硫肥可以改善作物生
  育性状，增加叶绿素含量，单株结果枝数、单株饱果数、百粒质量均有所增加，增产
  效果明显。",

  "output": "在花生以及小麦上的研究也表现出类似的结果，施用硫肥可以改善作物生
  育性状，增加叶绿素含量，单株结果枝数、单株饱果数、百粒质量均有所增加，增产
  效果明显。"

}
```

### 8. 文学

```
{

  "input": "画面中短衣帮们互相谈笑风生，穿长衫的则背着手傲气的走进酒店里，这一
  派景象十分鲜活灵动，富有生活气息。",

  "output": "画面中短衣帮们互相谈笑风生，穿长衫的则背着手傲气地走进酒店里，这
  一派景象十分鲜活灵动，富有生活气息。"

},
{

  "input": "有时他会陷入自我怀疑之中，哀叹"凡是一切顶小的顶平凡的生活事业，也不
  全是为我这样人而有的。我有的也许正是为人不屑要的。",

  "output": "有时他会陷入自我怀疑之中，哀叹"凡是一切小而平凡的生活事业，也不全
  是为我这样人所拥有的。我有的也许是别人不屑要的。""

}
```

Figure 11: Examples of CL$^2$GEC.

**9. 艺术学**

```
{

  "input": "单簧管继续演奏鸭子的主题，不同的是大提琴分奏变为三连音节奏型，在情
    绪上弥补了单簧管和大管的慢节奏，使其紧张性增加，作曲家想在此处在描写出鸭子
    跳入水中后动作变得灵活起来。",

  "output": "单簧管继续演奏鸭子的主题，而大提琴的分奏转变为三连音节奏型。这种
    变化在情绪上弥补了单簧管和大管的慢节奏，增加了音乐的紧张感。作曲家意图通过
    这一改变来描绘鸭子跳入水中后动作变得更加灵活的场景。"

},
{

  "input": "尖锐的音响效果与不和协因素的出现似乎在提醒和暗示战争带给人们的痛苦
    仍未散去。",

  "output": "尖锐的音响效果与不和谐因素的出现似乎在提醒和暗示战争带给人们的痛
    苦仍未散去。"

}
```

**10. 哲学**

```
{

  "input": "唯心主义将世界的本原归结为意识，这实质上模糊了世界本来的模样，那么
    既然唯心主义并没能使人正确的认识世界，那么唯心主义在历史长河中能够长久存在
    的真正原因需要揭露。",

  "output": "唯心主义将世界的本原归结为意识，这实质上模糊了世界本来的模样，既
    然唯心主义并没能使人正确地认识世界，那么唯心主义在历史长河中能够长久存在的
    真正原因仍需要探索。"

},
{

  "input": "既然意识之主体性在自身内部建构起来的对象世界终将归于抽象，意识也因
    此成为"非对象性的存在物"，即意识等于无。",

  "output": "由于意识的主体性在其内部构建的对象世界终将归于抽象，因此意识也被
    认为是"非对象性的存在物"，也就是说，意识等于无。"

}
```

Figure 12: Examples of CL$^2$GEC.

