# OpenReview forum: "CL2GEC: A Multi-Discipline Benchmark for Continual Learning in Chinese Literature Grammatical Error Correction"
_ICLR.cc/2026/Conference — ICLR 2026 Conference Withdrawn Submission_

### Official Review · Reviewer_a1hc · 2025-10-30

**Soundness:** 2
**Presentation:** 3
**Contribution:** 2
**Rating:** 2
**Confidence:** 4

**Summary:**

The paper introduces CL2GEC, the first benchmark and evaluation suite for continual learning in Chinese Grammatical Error Correction (CGEC) across 10 academic disciplines. The benchmark consists of 10,000 manually curated and annotated sentences, each associated with up to three human references, drawn from diverse domains such as law, science, and art. CL²GEC enables controlled assessment of catastrophic forgetting and cross-domain transfer using both standard GEC metrics and continual learning metrics adapted for task-sequence settings. The paper provides extensive empirical results for large language models (LLMs) under various adaptation and continual learning strategies, including sequential fine-tuning, LoRA, replay-based, and regularization-based continual learning methods.

**Strengths:**

**Comprehensive Academic Chinese GEC Benchmark**: The CL²GEC dataset is not only substantial in size (10,000 sentences) but also features rigorous multi-stage curation, leveraging both automatic detectors and in-domain human experts. The annotated data covered multiple research fields, which ensure the diversity of the benchmark.

**First Chinese GEC benchmark for continue learning**: To the best of my knowledge, this is the first work that study the scenario of continue learning for Chinese GEC. **However**, the target of continue learning for such a GEC task is ambiguous.

**Rich Experimental Settings**: The authors provide detailed experimental setups, including both random and semantically informed task sequences, ablation on replay buffer size, and systematic results across multiple LLMs and adaptation strategies.
In-depth Evaluation and Metrics: The use of both standard GEC metrics (Precision, Recall, F₀.₅) and continual learning-specific metrics (Backward Transfer, Average Task Performance) brings nuance to the empirical analysis. Equation formalizations are transparent and align with continual learning literature (see Page 4, loss and optimization formulations).

**Weaknesses:**

**Missing Annotation Principles**: The paper did not describe the annotation principles for GEC task. The grammatical error can be corrected in multiple ways. In most previous works, the principle of minimal edit is applied in the data annotation. Without annotation principles or annotation guidelines, the quality and the consistency of labeled data can not be ensured.

**Problematic Data Filtering** In section 3.2 Data Annotation, the authors described "Only sentences flagged consistently by all  6 grammatical error detectors are kept".  However, the sentences that are judged as erroneous sentences by all models are simple samples, while the difficult samples may be excluded by such a principle. This will significantly influence the distribution of the benchmark.

**Limited Research Significance for Continue Learning**: GEC task is not a long-context task and the dataset has only 10000 samples in total, which means the train cost on CL2GEC is not that unaffordable. Why not just shuffle the dataset randomly and directly train the model using all training data? The application of continue learning seems meaningless in such a task.

**Limited Theoretical Insights into Catastrophic Forgetting**: While the empirical results are substantial, there is little in-depth mathematical or theoretical exposition regarding why certain CL strategies succeed or fail in this linguistic, multi-domain context. For example, the paper lacks a formal analysis of error distribution shift or domain overlap between disciplines.

**Ambiguous Treatment of Semantic Task Ordering**: The computation of semantic similarity (Appendix A.1.2) is described, but the similarity is a metric for every pair of sub-dataset. The paper did not describe how to organize them into a absolute sorted list. From the Appendix A.1.2, I cannot know the specific rules for the 3 groups and how to sort them.

**Typos**: In line 462, there is reference error (Figure ??)

**Questions:**

- Could you please provide the annotation guidelines?
- How do you ensure the data quality?
- Could you please clarify the purpose of continue learning in GEC task? Can the continue learning reach a higher performance than direct training using all data?

---

> ### Author Response · Authors · 2025-12-03
>
> Thanks for your valuable and comprehensive comments. We attach great importance to your questions and hope to address your concerns.
>
> ---
>
> **Q1: Missing Annotation Principles: The paper did not describe the annotation principles for GEC task. The grammatical error can be corrected in multiple ways. In most previous works, the principle of minimal edit is applied in the data annotation. Without annotation principles or annotation guidelines, the quality and the consistency of labeled data can not be ensured.**
>
> A1:We focus on describing the multi-stage annotation pipeline that ensures high-quality data through various stages, including dual human annotation and expert review.
>
> We agree that making these principles more explicit in the paper would be helpful. In the revised version, we will (i) briefly summarize the annotation guidelines in the main text and (ii) include an inter-annotator consistency analysis based on the dual-annotation stage to further demonstrate the reliability of our labels.
>
> ---
>
> **Q2: Problematic Data Filtering In section 3.2 Data Annotation, the authors described "Only sentences flagged consistently by all 6 grammatical error detectors are kept". However, the sentences that are judged as erroneous sentences by all models are simple samples, while the difficult samples may be excluded by such a principle. This will significantly influence the distribution of the benchmark.**
>
> A2: We use six strong CGEC detectors (including GECToR and fine-tuned Chinese-BART). Being detected by all of them does not imply that the errors are simple; it implies that the sentence very likely contains at least one genuine error, which is crucial for cost-effective annotation given the size of the raw corpus.
>
> More importantly:
>
> - During human annotation + expert validation, annotators are required to correct all errors in the sentence, not only the ones highlighted by the detectors. Domain experts often introduce additional edits beyond the initial detector suggestions.
> - As a result, many final examples contain complex, multi-error corrections, despite originating from detector consensus.
> - At the dataset level, we also balance 10 primary disciplines with 1,000 sentences each, so no single domain dominates the distribution.
>
>  Therefore, although we start from consensus-filtered candidates, the final benchmark is not restricted to trivial errors and does not exhibit a biased or degenerate error distribution.
>
> ---
>
> **Q3: Limited Research Significance for Continue Learning: GEC task is not a long-context task and the dataset has only 10000 samples in total, which means the train cost on CL2GEC is not that unaffordable. Why not just shuffle the dataset randomly and directly train the model using all training data? The application of continue learning seems meaningless in such a task.**
>
> A3: We understand the reviewer’s point regarding the computational cost of joint training on the entire dataset. However, the primary value of CL in GEC is to simulate real-world scenarios where data is received sequentially and models must adapt without access to previous data. This is a scenario that is common in academic writing tasks, where new papers are continually submitted, but past data cannot be retained due to privacy, storage, and computational constraints.
>
> While full-data fine-tuning may provide better performance in the short term, CL methods allow for incremental learning without the need to retrain from scratch, which is particularly valuable in practice.

---

> ### Author Response · Authors · 2025-12-03
>
> **Q4: Limited Theoretical Insights into Catastrophic Forgetting: While the empirical results are substantial, there is little in-depth mathematical or theoretical exposition regarding why certain CL strategies succeed or fail in this linguistic, multi-domain context. For example, the paper lacks a formal analysis of error distribution shift or domain overlap between disciplines.**
>
> A4: Our primary goal in this work is to introduce CL^2GEC as a benchmark and to provide a comprehensive empirical study of CL methods for LLM-based CGEC. We appreciate the reviewer’s comment on the lack of theoretical analysis. We agree that deeper theoretical insights into catastrophic forgetting and error distribution shifts between domains would add significant value.
>
> In the revised version, we will include a more detailed discussion on how linguistic tasks and domain differences  affect CL performance. We will also explain why certain CL strategies perform better in GEC tasks, particularly with respect to domain overlap and error distribution shifts.
>
> ---
>
> **Q5: Ambiguous Treatment of Semantic Task Ordering: The computation of semantic similarity (Appendix A.1.2) is described, but the similarity is a metric for every pair of sub-dataset. The paper did not describe how to organize them into a absolute sorted list. From the Appendix A.1.2, I cannot know the specific rules for the 3 groups and how to sort them.**
>
> A5: We appreciate the opportunity to clarify this. In Appendix A.1.2, we explain that we compute an embedding for each discipline, derive a pairwise cosine similarity matrix, and group disciplines with high mutual similarity into three semantically coherent clusters. For the semantic task order, we first cluster disciplines into these three groups and then, within each group, order disciplines by their similarity to the previously trained task, so that the curriculum progresses with as smooth transitions as possible.
>
> We agree that the current text does not spell out this procedure clearly enough. In the revised version, we will explicitly describe how we construct the semantic task order, including both the grouping of disciplines and the within-group ordering.

---

### Official Review · Reviewer_nRwe · 2025-11-01

**Soundness:** 2
**Presentation:** 3
**Contribution:** 3
**Rating:** 2
**Confidence:** 3

**Summary:**

This paper presents the CL^2GEC dataset, which contains 10,000 high-quality, human-annotated samples for grammatical error correction, featuring a diverse range of error patterns. In addition, the paper conducts continual learning experiments on the CL^2GEC dataset and establish solid baselines for future work.

**Strengths:**

1. A large-scale, high-quality GEC dataset has been constructed.

2. Comprehensive analytical experiments are conducted to evaluate the performance of existing models.

**Weaknesses:**

1. The motivation for applying continual learning (CL) in the GEC domain is not clearly justified. Since grammatical errors across different domains of the same language share certain common patterns, it is unclear whether adopting a CL framework provides practical value.

2. As GEC dataset annotation is inherently challenging, the paper lacks consistency metrics to demonstrate the reliability of human annotations.

**Questions:**

1. As mentioned above, considering the commonality of error patterns in GEC, the necessity of CL in this domain still requires further validation. Adding a full-data fine-tuning result in Table 2 would make the argument more convincing.

2. Typo issue: there is a citation error between lines 43–46.

---

> ### Author Response · Authors · 2025-12-03
>
> Thanks for your valuable and comprehensive comments. We attach great importance to your questions and hope to address your concerns.
>
> ---
>
> **Q1: The motivation for applying continual learning (CL) in the GEC domain is not clearly justified. Since grammatical errors across different domains of the same language share certain common patterns, it is unclear whether adopting a CL framework provides practical value.**
>
> A1: The core motivation for applying CL in GEC is rooted in real-world academic writing environments, where models need to adapt to new domains sequentially while retaining previous knowledge. Although there are common error patterns across domains, the distribution of grammatical errors and writing styles varies significantly across academic disciplines.
> This is the core challenge that CL addresses: incremental learning under real-world constraints where models must continue to learn new tasks without forgetting previous knowledge. Our experiments show that catastrophic forgetting occurs even for large LLMs when sequentially adapting to new domains, and CL methods help reduce this forgetting, but they do not eliminate it entirely.
> We agree that a full-data fine-tuning baseline  would provide a useful upper bound. We will add this result to Table 2 in the revised version.
>
> **Q2: As GEC dataset annotation is inherently challenging, the paper lacks consistency metrics to demonstrate the reliability of human annotations.**
>
> A2: We focus on describing the multi-stage annotation pipeline, which is designed to ensure high-quality annotations, including dual human annotation and 100% expert validation by domain experts.
> We agree with the reviewer that, beyond this qualitative description, it is useful to report a quantitative consistency metric. In the revised version, we will add a inter-annotator consistency analysis based on the dual-annotation stage. This will provide a more explicit numerical indicator of annotation reliability on top of the described workflow.

---

### Official Review · Reviewer_tXfj · 2025-11-03

**Soundness:** 2
**Presentation:** 2
**Contribution:** 1
**Rating:** 2
**Confidence:** 4

**Summary:**

This paper introduces $CL^{2}GEC$, the first continual learning (CL) benchmark for multi-disciplinary academic writing. The benchmark comprises 10,000 human-annotated sentences from 10 academic fields, designed to evaluate models in a sequential, domain-incremental setting. The authors benchmarked large language models using sequential fine-tuning, parameter-efficient adaptation, and four CL algorithms.

**Strengths:**

1. This paper first proposes the Continual Learning benchmark for Chinese Literature Grammatical Error Correction, which could comprehensively evaluate the continual learning ability of LLMs in the chinese literature grammatical error correction task.

**Weaknesses:**

1. This paper only re-implement classic continual learning methods, which are not specifically designed for LLMs' continual learning. This paper should report the newest LLM continual learning methods[1,2] and provide convincing experiments to demonstrate the value of the dataset and the pros and cons of different methods.

2. It is unclear what its core differences from other datasets are. The dataset is built with grammatical error data from 10 different domains, but the connection between this domain categorization and continual learning is not immediately obvious.

3. As detailed in ICLR call for papers, the main text should be 9 pages or fewer, and additional pages are only allowed for the bibliography/references. Thus, the limitations should be controlled in 9 pages.

4. This paper only evaluates two common large language models, it fails to meet the acceptance standards for ICLR in terms of evaluation comprehensiveness, dataset indispensability.

[1] He, Jinghan, et al. "Continual instruction tuning for large multimodal models." arXiv preprint arXiv:2311.16206 (2023).

[2] Smith, James Seale, et al. "Coda-prompt: Continual decomposed attention-based prompting for rehearsal-free continual learning." Proceedings of the IEEE/CVF conference on computer vision and pattern recognition. 2023.

**Questions:**

Please refer to the weaknesses.

---

> ### Author Response · Authors · 2025-12-03
>
> Thanks for your valuable and comprehensive comments. We attach great importance to your questions and hope to address your concerns.
>
> ---
>
> **Q1: This paper only re-implement classic continual learning methods, which are not specifically designed for LLMs' continual learning. This paper should report the newest LLM continual learning methods[1,2] and provide convincing experiments to demonstrate the value of the dataset and the pros and cons of different methods.**
>
> A1:
>
> 1. Our main contribution is to introduce CL2GEC as a continual learning benchmark for multi-disciplinary Chinese academic GEC and to provide a systematic analysis of CL behavior on strong LLMs. We do not aim to propose a new CL algorithm in this paper.For this reason, we deliberately choose representative CL methods from the main families (regularization, replay, and knowledge distillation / orthogonal gradients: EWC, GEM, LwF, OGD), plus sequential fine-tuning and LoRA. These methods are architecture-agnostic, widely used in the CL literature (including in recent works on LLMs and LMMs), and well-understood in terms of their strengths and weaknesses. Using them allows us to:
>
>    - show that catastrophic forgetting is still substantial for LLMs in domain-incremental CGEC, and
>
>    - demonstrate that CL²GEC can clearly differentiate methods (e.g., regularization vs replay vs LoRA) in terms of forgetting, transfer, and stability.
>
>    This directly serves the benchmark’s purpose: to expose CL challenges and provide a reliable platform for future LLM-specific methods.
>
> 2. Although these CL algorithms were originally proposed for smaller models, they are not restricted to that setting. In our work, they are applied on top of instruction-tuned LLMs via parameter-efficient adaptation (e.g., LoRA-based updates), and the experiments show that:
>
>    - naive sequential fine-tuning cause strong forgetting across disciplines,
>    - the CL methods significantly reduce forgetting, but with different trade-offs.
>
>    This indicates that these methods are still meaningful baselines for LLM continual learning in the Chinese GEC scenario, and that CL²GEC is able to reveal their pros and cons.
>
> 3. We appreciate the reviewer’s pointers to [1] and [2], but both methods target settings that are fundamentally different from our text-only Chinese GEC scenario.
>    He et al. [1] study continual instruction tuning for multimodal models with an image encoder and vision–language tasks, while CODA-Prompt [2] is a rehearsal-free prompt-based method designed specifically for Vision Transformers on image classification.
>    Neither method is directly plug-and-play for decoder-only LLMs doing sequence-to-sequence grammatical error correction
>
> [1] He, Jinghan, et al. "Continual instruction tuning for large multimodal models." arXiv preprint arXiv:2311.16206 (2023).
>
> [2] Smith, James Seale, et al. "Coda-prompt: Continual decomposed attention-based prompting for rehearsal-free continual learning." Proceedings of the IEEE/CVF conference on computer vision and pattern recognition. 2023.
>
> ---
>
> **Q2: It is unclear what its core differences from other datasets are. The dataset is built with grammatical error data from 10 different domains, but the connection between this domain categorization and continual learning is not immediately obvious.**
>
> A2: Compared with existing Chinese GEC benchmarks, which are mostly general-domain and static, CL2GEC focuses on multi-disciplinary academic writing and is designed from the ground up for domain-incremental continual learning.We collect real sentences from 10 first-level disciplines in the Chinese subject taxonomy. In our CL setting, each discipline is explicitly treated as one CL task, and we provide a task sequence over these 10 domains to simulate the chronological arrival of manuscripts from different fields.
>
> This design allows us to systematically evaluate:
>
> * in-domain grammatical accuracy,
> * cross-discipline transfer,
> * resistance to catastrophic forgetting
>
> under realistic domain-incremental conditions—questions that existing static CGEC datasets cannot answer.

---

> ### Author Response · Authors · 2025-12-03
>
> **Q3: As detailed in ICLR call for papers, the main text should be 9 pages or fewer, and additional pages are only allowed for the bibliography/references. Thus, the limitations should be controlled in 9 pages.**
>
> A3: We thank the reviewer for pointing out the page-limit issue. We will revise the manuscript to ensure that the entire main text, including the Limitations section, fits within 9 pages, moving any excess details to the appendix.
>
> ---
>
> **Q4: This paper only evaluates two common large language models, it fails to meet the acceptance standards for ICLR in terms of evaluation comprehensiveness, dataset indispensability.**
>
> We appreciate the reviewer’s suggestion. We agree that more models would further strengthen the study, but we do not believe that dataset indispensability are questionable.
>
> * Our initial experiments focused on Qwen2.5-7B-Instruct and LLaMA3-8B-Instruct, which are both strong models for Chinese tasks and widely used in the community. We selected these models to demonstrate the effectiveness of CL methods across LLM architectures. We will extend our experiments to include additional Qwen2.5 variants (e.g., different sizes) in the final version to provide a broader evaluation of CL scalability across model sizes.
> * The “indispensability” of CL^2GEC lies in the type of questions it enables: existing static CGEC benchmarks cannot address whether *modern LLMs can retain grammatical knowledge while sequentially adapting to new scientific disciplines*. Our experiments show that catastrophic forgetting remains a substantial issue in this setting, and that CL methods help but do not fully solve it—insights that require a sequential, multi-domain benchmark like CL^2GEC.

---

### Official Review · Reviewer_cLmW · 2025-11-03

**Soundness:** 1
**Presentation:** 2
**Contribution:** 1
**Rating:** 2
**Confidence:** 4

**Summary:**

The paper introduces **CL2GEC**, the first continual learning benchmark for **Chinese Grammatical Error Correction (CGEC)** across academic disciplines. It studies how large language models adapt to new domains sequentially, highlighting the issue of **catastrophic forgetting**. While the benchmark assumes that models cannot perform **multi-task training** or access all domains simultaneously, this constraint may be unrealistic for modern LLMs that already demonstrate strong **cross-domain generalization and in-context learning**. Experiments on Qwen2.5 and LLaMA3 show that **regularization-based continual learning methods** outperform naive fine-tuning and replay strategies. Overall, CL2GEC provides a valuable research framework for studying lifelong adaptation, though its sequential learning assumption may limit its real-world applicability.

**Strengths:**

1. The paper introduces a **novel multi-disciplinary Chinese Grammatical Error Correction (GEC) dataset** with a **comprehensive and well-structured evaluation framework**.

2. Experiments on Qwen2.5 and LLaMA3 show the proposed method outperforms naive fine-tuning and replay strategies.

**Weaknesses:**

1. The paper assumes that large language models (LLMs) can only acquire multi-domain GEC capabilities through **continual learning**, without comparing other plausible approaches such as **multi-task fine-tuning, retrieval-augmented generation (RAG), or in-context learning**. Given the strong generalization ability of LLMs, continual learning may not be strictly necessary in this setting.

2. The model comparison is limited; it should include **more open-source models with stronger Chinese capabilities**, such as **different sizes of the Qwen2.5 series**, to provide a fairer evaluation.

3. The study **does not compare smaller encoder–decoder models**, for which continual learning might actually be **more relevant and effective** than for large instruction-tuned LLMs.

**Questions:**

N/A

---

> ### Author Response · Authors · 2025-12-03
>
> Thanks for your valuable and comprehensive comments. We attach great importance to your questions and hope to address your concerns.
>
> ---
>
> **Q1: The paper assumes that large language models (LLMs) can only acquire multi-domain GEC capabilities through continual learning, without comparing other plausible approaches such as multi-task fine-tuning, retrieval-augmented generation (RAG), or in-context learning. Given the strong generalization ability of LLMs, continual learning may not be strictly necessary in this setting.**
>
> A1: We would like to clarify that CL is not the only solution. In our study, we focus on sequential adaptation in scenarios where historical data cannot be accessed due to privacy, storage, and annotation constraints. This is a real-world problem in many academic and professional settings, where data arrives incrementally.
>
> > **Original Paper Quote**: *"Continual Learning (CL) addresses the challenge of learning from a stream of tasks {D1, . . . , DN} without access to previous task data"*
> >
> > **Original Paper Quote**: *"NLP tasks  has received almost no attention in CGEC, leaving an open question: Can modern LLMs retain grammatical knowledge while sequentially adapting to new scientific disciplines?"*
>
> The key point of our research is to study how CL mitigates catastrophic forgetting in such constrained settings, not to claim that CL is the only possible approach. While we recognize the value of multi-task and ICL methods, we argue that CL is crucial when joint training is not feasible due to the reasons mentioned.
>
> ---
>
> **Q2: The model comparison is limited; it should include more open-source models with stronger Chinese capabilities, such as different sizes of the Qwen2.5 series, to provide a fairer evaluation.**
>
> A2: We appreciate the reviewer’s suggestion. Our initial experiments focused on Qwen2.5-7B-Instruct and LLaMA3-8B-Instruct, which are both strong models for Chinese tasks and widely used in the community. We selected these models to demonstrate the effectiveness of CL methods across LLM architectures.
> We will extend our experiments to include additional Qwen2.5 variants (e.g., different sizes) in the final version to provide a broader evaluation of CL scalability across model sizes.
>
> ---
>
> **Q3: The study does not compare smaller encoder–decoder models, for which continual learning might actually be more relevant and effective than for large instruction-tuned LLMs.**
>
> A3: We agree that smaller encoder-decoder models may benefit more from CL due to their smaller capacity to retain knowledge across tasks. However, our primary focus in this work was to study LLMs due to their growing prevalence in real-world academic settings.
> While CL^2GEC is model-agnostic and can easily be extended to encoder-decoder models (like Chinese-BART and GECToR, which are already used in our pipeline for error detection), we chose to focus on LLMs first because they present real challenges with catastrophic forgetting when trained sequentially as we can see in table 2

---

### Official Review · Reviewer_2STk · 2025-11-04

**Soundness:** 3
**Presentation:** 3
**Contribution:** 2
**Rating:** 6
**Confidence:** 3

**Summary:**

The paper proposes CL2GEC which is a benchmark for Chinese Literature Grammatical Error Correction across multiple disciplines. In particular, CL2GEC is designed to support continual learning. The paper explores various continual learning algorithms on this dataset, and the authors perform various experiements to highlight important dynamics (e.g., task ordering, backbone differences) that practitioners should be aware of. Evaluation is done using both standard GEC metrics, as well as continual learning metrics.

## Dataset
Dataset was crawled from China National Knowledge Infrastructure, with 10 disciplines: Law, Management, Education, Economics, Science, History, Agriculture, Literature, Art, and Philosophy. Random sample of 1000 questions per discipline to form 10k questions in total. Then some cleaning steps are done, such as sentence extraction, noise removal, and anonymization. Finally, the data is annotated both by LLMs (initial filter), then manually reviewed by expert annotators.

**Strengths:**

- baselines are thorough
- it's a big effort to collect 10,000 human annotated sources, so this is a valuable resource
    - Thorough data collection process, with manual human reviews, so the dataset is likely high quality
- experiments had good coverage of various methods.
- ablations are interesting. For example, I liked the section on task order

**Weaknesses:**

- This is a very narrow domain and not easily generalizable to some of the bigger topics that the community really cares about. I imagine the subset of researchers who care about Chinese Literature GEC might not be that large.
- I would have wanted the authors to flesh out more what makes this task special from other GEC tasks. Are there any nuances specific to this task that are less common in other GEC tasks?
- I don't fully see how this dataset itself is connected to continual learning. It somehow feels like the authors just stitched two somewhat disjointed topics together (GEC + continual learning).

**Questions:**

- How much or how little did you ablate on the different filtering steps? For example, in the "Noise Removal" step, did you iterate much on the parameters here? Or did you just use standard reasonable assumptions? I'm quite curious on how some of these filtering parameters may affect the performance.
- curious on how this will transfer to other architectures (e.g. state sapace models), and also other model sizes (No need for extra experiments! Just curious if this is something you've done)
- How do you ensure annotators are annotating with the same metric/criteria in mind?

---

### Note · Authors · 2025-12-04

I have read and agree with the venue's withdrawal policy on behalf of myself and my co-authors.